# Entrainment and the tropical tropospheric thermal structure in global climate models

Lucinda A. Palmer<sup>1</sup> and Martin S. Singh<sup>2,3</sup>

<sup>1</sup>School of Earth and Environmental Sciences, University of St Andrews, St Andrews, UK.

<sup>2</sup>School of Earth, Atmosphere & Environment, Monash University, Victoria, Australia.

<sup>3</sup>Centre of Excellence for Weather in the 21st Century, Monash University, Victoria, Australia.

**Correspondence:** Lucinda A. Palmer (lap20@st-andrews.ac.uk)

Abstract. The observed relationship between stability and humidity in the tropical troposphere has been argued to be strongly influenced by moist convective entrainment (Palmer and Singh, 2024). In this study, we investigate this relationship in fourteen models from phase 6 of the Coupled Model Intercomparison Project with the aim of evaluating their representation of such entrainment processes. We define a diagnostic of convective entrainment using the climatological slope of the relationship between measures of lower-tropospheric stability and humidity in precipitating regions of the tropics. While some models reproduce the sign of this slope as estimated from reanalyses, others produce weak or opposing relationships between stability and humidity, implying unphysical entrainment rates. We relate these contrasting behaviours to aspects of the models' convection schemes; models that employ plume-based cloud models and traditional "CAPE" closures, where convection is assumed to remove cloud buoyancy over a specified timescale, tend to better reproduce reanalyses. We also explore the use of the stability-humidity relationship to constrain projections of extremes in convective available potential energy (CAPE) and boundary-layer moist static energy (MSE). These quantities have been argued to be influenced by convective entrainment and are relevant to intense thunderstorms and humid heatwaves, respectively. We find that models that quantitatively reproduce the stability-humidity relationship in reanalyses tend to produce higher increases in CAPE and boundary-layer MSE under warming. However, due to observational uncertainties and model scatter, further work is required to develop a strong emergent constraint.

## 1 Introduction

The thermal structure of the tropical atmosphere is controlled by processes ranging from cloud microphysics to atmospheric dynamics that act on a wide range of spatial scales (Riehl and Malkus, 1958; Xu and Emanuel, 1989; Singh and O'Gorman, 2013; Bao and Stevens, 2021; Williams et al., 2023). Accurately representing the distribution of temperature within the troposphere in a global simulation is therefore a strong test of model performance. Moreover, changes to the tropical thermal structure under warming have been shown to have important implications for convective storm intensity (Singh et al., 2017), humid heatwaves (Duan et al., 2024), and radiative feedbacks (Ceppi and Gregory, 2017), emphasising the importance of accurately representing the tropical thermal structure in climate projections.

A process of particular importance in setting the thermal structure of the tropics is that of moist convection (Betts, 1982; Xu and Emanuel, 1989; Emanuel, 2007; Bao and Stevens, 2021). A number of recent papers have highlighted the effect of convective entrainment on the static stability within the tropics (e.g., Singh and O'Gorman, 2013; Miyawaki et al., 2020; Bao et al., 2021). In particular, Palmer and Singh (2024) argued that the relationship between tropospheric humidity and the static stability within regions of rainfall may be used to estimate an effective bulk entrainment rate for moist convection. In this paper, we apply the analysis of Palmer and Singh (2024) to climate models participating in the Phase 6 of the Coupled Model Intercomparison Project (Eyring et al., 2016, CMIP6). Our aims are two fold: (1) to determine whether relationships between stability and humidity in the tropical troposphere may be used to analyse the representation of moist convective entrainment in climate models, and (2) to determine whether such relationships, when compared to observational analyses, may be used to constrain projections of extremes hypothesised to be sensitive to entrainment—namely the potential for intense thunderstorms and humid heatwaves.

Our approach is based on the hypothesis that convection adjusts the atmospheric thermal structure to be close to neutrally buoyant to rising air within clouds (e.g., Arakawa and Schubert, 1974). Due to the effect of entrainment, this allows the lapse rate in convecting regions to be maintained as less stable than that of a moist adiabat (Singh and O'Gorman, 2013). Further, since the magnitude of the effect of entrainment depends on the environmental humidity, this suggests that moister regions would tend toward higher stability relative to drier regions, provided they are sufficiently close to moist convection (Singh and Neogi, 2022; Singh et al., 2019). Although horizontal gradients in temperature are also constrained by large-scale dynamics (Romps, 2021; Bao et al., 2022), Palmer and Singh (2024) found relationships between humidity and stability on daily scales in reanalysis that are consistent with the above-described hypothesis, raising the possibility of constraining moist convective entrainment through analysis of large-scale fields alone.

35

Our analysis finds a wide range of stability-humidity relationships in CMIP6 models, supporting the argument that they are determined by uncertain convective physics and allowing us to diagnose a measure of convective entrainment that we show to to be related to structural choices within models' convection schemes. Emmenegger et al. (2024) recently used similar arguments to derive an analogousdiagnostic they termed "pseudo-entrainment" that is based on the mean stability and humidity in convecting regions rather than their spatial and temporal variability. We also compare our results to those of Ahmed and Neelin (2021), who developeddiagnostics targeted at convective physics based on the relationship between precipitation and plume buoyancy established by Ahmed and Neelin (2018). The authors found that some climate models have a precipitation-buoyancy relationship similar to reanalysis, whilst precipitation in other models is either inadequately sensitive or too sensitive to humidity in the lower free troposphere.

Along with being a key model parameter, convective entrainment has been argued to play a role in modulating impacts of climate change, including affecting future increases in the potential for intense thunderstorms (Singh et al., 2017) and the frequency and intensity of humid heatwaves (Duan et al., 2024). Therefore, we also explore how the diagnostics we define may be able to help constrain climate model projections of these important extremes under warming. Specifically, we investigate how CMIP6 projections of extremes in convective available potential energy (CAPE), representing intense thunderstorm potential, and near-surface moist static energy, representing a measure of the intensity of humid heatwaves, relate

to the climatological relationship between stability and humidity across the model ensemble. On average, models that reproduce the stability-humidity relationship found in reanalysis show higher increases in CAPE and the intensity of humid heatwaves under warming, but uncertainties remain relating to the sufficiency of reanalysis to provide the observational constaint, and the interpretation of the model scatter through the lens of convective entrainment.

Our findings are organised as follows. We first introduce a method for evaluating the stability-humidity relationship and describe the theory relating it to convective entrainment in Section 2. In Section 3 we examine the CMIP6 ensemble from this perspective and the varied ability of models to reproduce observed relationships. We then investigate the implications of these results, using humid heatwaves and extreme CAPE as examples in Section 4. Lastly, in Section 5, we conclude and suggest avenues for further research.

## 2 Entrainment and the stability-humidity relationship

80

To begin, we follow Palmer and Singh (2024) and derive a simple model for the relationship between stability and humidity in convective regions. The key assumption, known as the zero-buoyancy plume (ZBP) assumption, is that convection maintains a lapse rate that is neutrally buoyant with respect to an entraining plume. Under these conditions and neglecting the effect of water vapour on density, Singh and O'Gorman (2013) showed that the saturation moist static energy (MSE\*) of the free troposphere is governed by

$$\frac{d\text{MSE}^*}{dz} = -\epsilon \ell (q^* - q),\tag{1}$$

where  $\mathrm{MSE}^* = c_p T + gz + \ell q^*$ ,  $\epsilon$  is the entrainment rate,  $\ell(q^* - q)$  is the saturation deficit (expressed in energy units) and z is height. Here,  $c_p$  is the isobaric specific heat capacity of air,  $\ell$  is the latent enthalpy of vaporisation, T is the temperature, q is the specific humidity,  $q^*$  is the saturation specific humidity, and g is the gravitational acceleration.

Following Palmer and Singh (2024) we focus on the region of the troposphere between 850 hPa and 500 hPa. We integrate (1) in height, assuming the entrainment rate and latent enthalpy of vaporisation are constant, giving

$$\Delta MSE^* = -\epsilon \Delta z \ell \overline{(q^* - q)}$$
 (2)

where  $\Delta \text{MSE}^*$  is the change in saturation moist static energy between 850 hPa and 500 hPa,  $\Delta z$  is the height difference between these pressure levels, and

$$\overline{(q^* - q)} = \frac{1}{\Delta z} \int_{z_{250}}^{z_{500}} (q^* - q) \, dz,$$

is the height-weighted mean saturation deficit. Eq. (2) implies that the vertical change in MSE\* is related to the average free-tropospheric saturation deficit, where the change in MSE\* is a measure of stability, and saturation deficit is a measure of humidity.

Solutions of (2) are plotted on Fig. 1 for three different entrainment rates of 0.1, 0.25 and 0.5 km<sup>-1</sup> and for  $\Delta z = 4.36$  km, equal to its mean value within 20°S–20°N in the 5th generation European Centre for Medium-Range Weather Forecasts Atmospheric reanalysis (Hersbach et al., 2020, ERA5) for the years 2000-2020. Overlaid are 2D histograms of  $\Delta$ MSE\* and

Figure 1. 2D histograms depicting the relationship between the daily-mean vertical difference in MSE\* and vertically-averaged saturation deficit  $\ell(\overline{q^*-q})$  calculated between 850 hPa and 500 hPa for ocean (a,b) and land (c,d) in ERA5. (a, c) include both precipitating and non-precipitating points and (b,d) include only points that exceed 5 mm day<sup>-1</sup> of precipitation. Black lines give solutions to (2) for entrainment rates as given in (a).

the saturation deficit  $\ell(q^* - q)$  calculated using daily-mean fields taken from ERA5 for the same years and in the region 20°S–20°N over ocean (Fig. 1a, b) and land (Fig. 1c, d). Since the ZBP assumption is only applicable to convecting regions, we also plot histograms limited to locations with at least 5 mm day<sup>-1</sup> of precipitation (Fig. 1b, d).

As found by Palmer and Singh (2024), the histograms show a clear relationship between stability and humidity, particularly when only convective regions are considered. Comparing to the theoretical solutions, the relationship suggests an effective entrainment rate in the range 0.15-0.4 km<sup>-1</sup>. Over land, the implied slope is steeper, corresponding to a higher entrainment rate. This is in contrast to previous studies which suggest that entrainment tends to be lower over land compared to over the ocean (Lucas et al., 1994; Kirshbaum and Lamer, 2021; Takahashi et al., 2023). One possible explanation for this is that the surface and boundary layer height is larger and more variable over land. If the boundary layer extends above 850 hPa, part of the profile may be better modelled as a dry adiabat, resulting in a large value of instability. This would more likely occur over drier

**Figure 2.** The ellipse fitting process on ERA5 using the OpenCV Python library. The shaded blue region shows the original histogram for precipitating points in ERA5 over tropical ocean as in Fig. 1b. Countours of frequency are plotted enclosing 30% (inner grey), 75% (black), and 90% (outer grey) of the total histogram. The red ellipse is the fitted ellipse to the 75% contour and the red line passing through it is the semi-major axis extended to the axes limits.

surfaces, where the lifting condensation level is high, potentially resulting in a correlation with the free-tropospheric humidity unrelated to variations in entrainment. Further, since we consider only daily averages, variations in humidity or boundary-layer height on sub-daily timescales may also affect our interpretation of the stability-humidity relationship over land regions. However, we found the boundary layer height, as diagnosed by ERA5, is rarely above 850 hPa and so is unlikely to account for the larger slope of the relationship over land regions, at least not in a straightforward way. Understanding the difference between land and ocean is of interest, but here we focus on the troposphere over the ocean, where any interference from the boundary layer is minimal.

Thus far, the relationship between stability and humidity, and any inference regarding convective entrainment, has been evaluated by eye, but a method to objectively quantify it is needed to make the comparison between ERA5 and CMIP6. To do this, we estimate the slope and 'strength' of the relationship between stability and humidity based on the histogram. Specifically, we draw a contour of constant frequency on the histogram that encapsulates 75% of the data for precipitating points only over tropical ocean. An ellipse is then fitted to the contour using the OpenCV Python library (Itseez, 2015) that allows an ellipse to be fitted to a binary image. We then identify the slope of the semi-major axis of this ellipse as  $\epsilon_d/\Delta z$ , where  $\epsilon_d$  is the diagnosed entrainment rate and  $\Delta z$  is the mean height difference between 850 hPa and 500 hPa. We further take the ratio between the length of the semi-major and semi-minor axes as a measure of the strength of the relationship, where a larger axis ratio is indicative of a stronger relationship. Figure 2 shows an example of this calculation for the ERA5 reanalysis. The resultant diagnosed entrainment rate is  $\epsilon_d = 0.154$  km<sup>-1</sup>.



As a rough measure of the structural uncertainty in estimating  $\epsilon_d$ , we redo the ellipse-fitting calculation using contours that encapsulate different fractions of the data from 30% to 90% in steps of one percent. For the ERA5 reanalysis, this gives a range

of diagnosed entrainment rates between 0.148 and 0.158 km<sup>-1</sup>, indicating that for ERA5, the estimation procedure is relatively robust.

## 3 The simulated stability-humidity relationship





Having developed a method to quantify the stability-humidity relationship in ERA5, we can apply it to CMIP6 models. Note that, when diagnosed in a model,  $\epsilon_d$  may be different to the actual entrainment rate used in the convection scheme, which itself may vary with environmental conditions. As we will see,  $\epsilon_d$  may not refer to a physically realisable entrainment rate at all. Rather, the simulated stability-humidity relationship may result from compensating errors in different components of the model so as to reproduce aspects of the observed distribution, rather than directly representing a physical process. Nevertheless, we will argue that the stability-humidity relationship can provide insights to model behaviour.

We select fourteen CMIP6 models for our analysis based on the availability of variables required. For each model, we consider the region 20°S–20°N over ocean and for the years 2000-2014 in a single ensemble member of the historical scenario. Noting too that precipitation values vary between models, we transition from identifying convective regions with a 5 mm day<sup>-1</sup> threshold to identifying convective regions with a threshold given by the 75th percentile of precipitation including all grid points in the region 20°S–20°N. The 75th percentile of precipitation was selected because it corresponds roughly to 5 mm day<sup>-1</sup> in ERA5.

To better quantify the uncertainty associated with observational estimates of the stability-humidity relationship, we follow Palmer and Singh (2024) and consider a second reanalysis, version 2 of the Modern-Era Retrospective analysis for Research and Applications (Gelaro et al., 2017, MERRA2). Both reanalyses produce similar histograms in the stability-humidity phase plane, although MERRA2 produces a slightly steeper slope between the two variables (Fig. 3a,b).

Compared to the two reanalyses, highly varying relationships between stability and the saturation deficit emerge in the fourteen CMIP6 models (Fig. 3). A clear negative slope, as found in ERA5 and MERRA2 and indicating a positive diagnosed entrainment rate, can be seen for around half the models. In some cases, the relationship is both stronger and has a steeper slope than in the reanalyses (e.g., Fig. 3c,d). However, some models depict little to no relationship between stability and humidity (Fig. 3k-m) and others have a relationship opposite to the reanalyses, in which stability increases with saturation deficit (Fig. 3n-p). It is clear in the latter case that other factors beyond convective entrainment are influencing the resulting relationship between stability and humidity.

We can summarise the above relationships by calculating the diagnosed entrainment rate  $\epsilon_d$  (Fig. 4) and plotting it against the axis ratio for each model and the reanalyses calculated analogously to that described for ERA5 above (Fig. 5). MERRA2 and ERA5 have very similar diagnosed entrainment rates, but the MERRA2 distribution contains slightly more spread, and therefore a lower axis ratio as well as a larger uncertainty in  $\epsilon_d$ , measured as the range produced when different contours are used to estimate it.

Eight of the models have robustly positive diagnosed entrainment rates, with the diagnosed entrainment rates ranging from substantially smaller to substantially larger than those estimated from reanalyses. However, for a number of models, there is

**Figure 3.** 2D histograms depicting the historical relationship between the vertical change in MSE\* and the saturation deficit between 850 hPa and 500 hPa for (a) ERA5, (b) MERRA2 and (c-p) fourteen CMIP6 models. The CMIP6 models are ordered from the largest negative slope to the largest positive slope. The distribution includes only ocean surface points that exceed the precipitation threshold between 20°N-20°S.

**Figure 4.** Diagnosed entrainment rate based on the 75% contour for each CMIP6 model. Error bars show the range of the diagnosed entrainment rate using contours ranging from 30% to 90%. Shading represents the convective schemes used by the models, either a CAPE closure (dark blue) or a non-CAPE closure (light blue), and if the model does not employ a plume (hatching). Vertical bars show diagnosed entrainment rate ranges for ERA5 (black) and MERRA2 (grey).

some sensitivity of the diagnosed entrainment rate to the particular contour chosen to estimate it (error bars in Fig. 4 & 5). As a result, all but two models with a positive diagnosed entrainment rate have a range of entrainment rates that overlap with at least one reanalysis.

There are six models that have either a very weak relationship between stability and humidity or a relationship in which stability increases with increasing saturation deficit. These models have a negative diagnosed entrainment rate (table 1 provides the value of  $\epsilon_d$  for each model). The axis ratio also varies considerably across models. There is not a strong relationship between the axis ratio and entrainment rate, however, larger magnitudes of  $\epsilon_d$  (both positive and negative) tend to be associated with larger axis ratios.

Figure 5. The diagnosed entrainment rate and the axis ratio (see text), representing the strength of the stability-humidity relationship, for ERA5, MERRA2 and each CMIP6 model as given in the legend. Error bars show the structural uncertainty (see Fig. 4) in the diagnosed entrainment rate; ERA5 error bars are too small to be visible. The value of the correlation coefficient, r, is calculated across models with positive diagnosed entrainment rates only.

#### 3.1 The diagnosed entrainment rate as a process-oriented diagnostic



Given the variation in the stability-humidity relationship across models, the diagnosed entrainment rate acts a useful diagnostic for evaluating model performance; both reanalyses agree well on the value of the diagnosed entrainment rate (Fig. 5), and it has a clear physical interpretation. For models for which stability decreases with saturation deficit, the diagnosed entrainment rate is positive, and it may be interpreted as a bulk measure of convective mixing in the lower troposphere. Differences in the value of the diagnosed entrainment rate between this subset of models and the reanalyses suggests that convective mixing in the simulations is either too weak or too strong. For models with negative diagnosed entrainment rates, however, this interpretation is no longer applicable; these models do not conform to expectations from the ZBP theory, and it is less clear what, if any, physical significance  $\epsilon_d$  carries.

An obvious question is whether differences in convection schemes can help explain the variety of stability-humidity relationships found in our study. We therefore collate the details of the convective scheme for each model in Table 1. We focus on three key aspects of convection schemes. The cloud model, which determines the vertical structure of the convective mass flux and its effect on grid-scale variables, the trigger, which determines when the convection scheme is active, and the closure, which determines the overall magnitude of the convective mass flux.

Table 1. Diagnosed entrainment rate  $\epsilon_d$  and details of the deep convective schemes employed by the fourteen CMIP6 models used in this study. The "q sensitivity" is the sensitivity to lower-tropospheric humidity diagnosed by Ahmed and Neelin (2021). Convection schemes are categorised by structural characteristics of their cloud model, trigger, and closure. Cloud models may use a single bulk entraining plume (bulk-plume), an ensemble of entraining plumes (multi-plume), or a non plume-based structure. The trigger may depend on the properties of an undiluted or diluted parcel ascent in the free troposphere (parcel), properties in the boundary layer (BL), the grid-scale relative humidity (RH), or dynamic variables on the grid-scale such as vertical velocity (w) or moisture convergence  $(\nabla \cdot \mathbf{u}q)$ . The convective closure calculates the cloud-base mass flux and may be a CAPE closure, which assumes that a measure of integrated cloud buoyancy is removed by convection over a specified timescale, a quasi-equilibrium closure, which assumes the convective tendency of integrated cloud buoyancy balances that of large-scale processes  $(\partial_t \text{CAPE})$ , or a closure based on assumptions about boundary-layer turbulence (BL) or convective adjustment. Two closures include prognostic equations from which the cloud-base mass flux is calculated.

| Model             | $\epsilon_d$ | q sensitivity | Cloud model     | Trigger                          | Closure                    | References                                                                      |
|-------------------|--------------|---------------|-----------------|----------------------------------|----------------------------|---------------------------------------------------------------------------------|
| KACE-1-0-G        | +0.357       | adequate      | bulk-plume      | parcel & BL                      | CAPE                       | Gregory and Rowntree (1990); Fritsch and Chappell (1980); Walters et al. (2019) |
| CNRM-CM6-1        | +0.280       | adequate      | bulk-plume      | parcel                           | CAPE                       | Guérémy (2011); Piriou et al. (2018);<br>Roehrig et al. (2020)                  |
| TaiESM1           | +0.246       | excessive     | multi-plume     | parcel & BL                      | CAPE                       | Zhang and McFarlane (1995); Neale et al. (2013); Wang et al. (2015)             |
| CNRM-CM6-1-<br>HR | +0.231       | adequate      | bulk-plume      | parcel                           | CAPE                       | Guérémy (2011); Piriou et al. (2018);<br>Roehrig et al. (2020)                  |
| MRI-ESM2-0        | +0.146       | adequate      | multi-plume     | parcel                           | CAPE                       | Yoshimura et al. (2015)                                                         |
| NorESM2-LM        | +0.124       | adequate      | multi-plume     | parcel                           | CAPE                       | Zhang and McFarlane (1995); Neale et al. (2013)                                 |
| GFDL-CM4          | +0.098       | excessive     | bulk-plume      | parcel & RH                      | CAPE                       | Bretherton et al. (2004); Zhao et al. (2018)                                    |
| MIROC6            | +0.062       | adequate      | multi-plume     | parcel                           | prognostic                 | Chikira and Sugiyama (2010); Ando et al. (2021)                                 |
| BCC-CSM2-MR       | -0.002       | excessive     | bulk-plume      | parcel & BL & $\boldsymbol{w}$   | $\partial_t \mathrm{CAPE}$ | Wu (2012); Wu et al. (2019)                                                     |
| CAS-FGOALS-g3     | -0.004       | excessive     | multi-plume     | parcel & RH                      | $\partial_t \mathrm{CAPE}$ | Zhang and McFarlane (1995); Zhang and Mu (2005)                                 |
| CCCma-CanESM5     | -0.033       | excessive     | multi-plume     | parcel                           | prognostic                 | Zhang and McFarlane (1995); Scinocca and McFarlane (2004)                       |
| IPSL-CM6A-LR      | -0.079       | excessive     | episodic mixing | BL                               | BL                         | Emanuel (1991); Rio et al. (2009); Rochetin et al. (2014)                       |
| MPI-ESM1-2-LR     | -0.178       | insufficient  | bulk-plume      | BL & $\nabla \cdot \mathbf{u} q$ | CAPE                       | Tiedtke (1989); Nordeng (1994); Möbis and Stevens (2012)                        |
| INM-CM5-0         | -0.256       |               | adjustment      | parcel                           | adjustment                 | Betts (1986)                                                                    |

Notably, the two models with the most pronounced positive slopes between stability and saturation deficit (indicating  $\epsilon_d < 0$ ), INM-CM5-0 and IPSL-CM6A-LR, have cloud models that structurally differ from the other CMIP6 models included in this study. While other models are generally based on either a bulk entraining plume or an ensemble of entraining plumes with distinct entrainment characteristics (e.g., Zhang and McFarlane, 1995; Arakawa and Schubert, 1974), INM-CM5-0 uses the convective adjustment scheme of Betts (1986), which has no explicit parameterisation of mixing, and IPSL-CM6A-LR uses a convection scheme that includes episodic mixing through a buoyancy sorting mechanism (Emanuel, 1991). For these two models, the form of the cloud model provides a physical reason for their lack of a decrease in stability with saturation deficit, and our results suggest deficiencies in the ability of these schemes to provide sufficient convective mixing.

That convection schemes without an entraining plume cannot reproduce a mechanism that is based in entrainment mixing is physically plausible; but what of other models that have negative or very small diagnosed entrainment rates? Examination of Table 1 reveals that similar cloud models can lead to both negative and positive  $\epsilon_d$  values. For example, four models (TaiESM1, NorESM2-LM, CAS-FGOALS-g3, CCCma-CanESM5) are based on the multi-plume representation of convection in Zhang and McFarlane (1995) and yet producemarkedly differing values of  $\epsilon_d$  of both signs. On the other hand, the convective closures do appear to differ systematically between models with positive and negative  $\epsilon_d$  values. Except for MIROC6, which has the smallest positive value of  $\epsilon_d$  across the ensemble, all other models for which  $\epsilon_d > 0$  have so-called "CAPE" closures. Further, only one model with  $\epsilon_d < 0$  (MPI-ESM1-2-LR) has such a closure.

A CAPE closure calculates the cloud-base mass flux by assuming a relation of the form,






$$\frac{\partial}{\partial t} \left( \int_{B>0} B \, dz \right)_{\text{conv}} = -\frac{1}{\tau} \left( \int_{B>0} B \, dz \right) \tag{3}$$

where B is a measure of cloud buoyancy and  $\tau$  is a timescale, typically on the order of a few hours. Here integrals are taken over heights at which the buoyancy is positive, and the left-hand side represents the rate of change due to the convection scheme. If B is the buoyancy of an adiabatic parcel, the integral in (3) is the CAPE. However, the models considered in this study all use a dilute ascent that includes entrainment in their calculation of buoyancy, and the integral may then be thought of as "dilute CAPE". Eq. (3) thus states that convection acts to rapidly remove buoyancy and relax the atmosphere towards a profile that is neutral to dilute ascent. This then provides a possible physical explanation for the differences in sign of  $\epsilon_d$  across the models; convection in models with CAPE closures rapidly equilibrates the atmosphere to neutral buoyancy, consistent with the ZBP assumption, resulting in a strong stability-humidity relationship. Other models have closures that balance the tendencies of convection with those of the large-scale ( $\partial_t$ CAPE), consider aspects of the boundary layer (BL), or use more complicated prognostic formulations. Evidently, these closures do not produce a strong stability-humidity relationship.

While the above argument provides a straightforward explanation for the difference between models with positive and negative diagnosed entrainment rates, some caveats must be noted. Firstly, we have compared our results to reanalyses, which may be influenced by the very convection schemes we are attempting to validate. In fact, both ERA5 and MERRA2 are based on atmospheric models that include plume-based convection schemes, and ERA5 includes a CAPE closure (ECMWF, 2016) while MERRA2 does not (Molod et al., 2015). We note, however, that Palmer and Singh (2024) found a negative slope

between stability and saturation deficit in estimates of these quantities from radiosonde profiles, providing at least qualitative direct observational confirmation for a positive diagnosed entrainment rate in the tropics.

A second caveat is that our argument cannot explain the results from MPI-ESM1-2-LR, which has a CAPE closure and a plume-based cloud model, and yet still produces a negative diagnosed entrainment rate. One possibility for this may be the criterion of moisture convergence in the convective trigger of MPI-ESM1-2-LR; only this model and BCC-CSM2-MR include grid-scale dynamical variables in their trigger function. Emmenegger et al. (2024) found that models with triggers that depend on moisture convergence have higher mean stability compared to observations. However, it is unclear why such a convective trigger would cause stability to have a weak dependence on humidity. The relationship between stability and humidity in models may therefore depend on more complex and possibly interwoven mechanisms that include both parameterisations and the dynamical core. Further work in which convection parameterisations are systematically altered would be useful to better understand how convection controls the stability-humidity relationship.







Finally, we compare our results to two recent studies (Ahmed and Neelin, 2021; Emmenegger et al., 2024) that have developed diagnostics similar to the diagnosed entrainment rate based on the finding that the buoyancy of an entraining plume acts as a good predictor for the onset of precipitation in observations (Ahmed and Neelin, 2018). By applying this idea to climate models, the authors were able to estimate how sensitive precipitation in the models was to humidity in comparison to ERA5.

Ahmed and Neelin (2021) classified 24 CMIP6 models (including 13 of the 14 considered here) as being either adequately sensitive, inadequately sensitive, or overly sensitive to a measure of plume buoyancy that depends strongly on lower-tropospheric humidity (Table 1). The physical basis for this sensitivity is the dilution of clouds by dry tropospheric air, and the authors find that the sensitivity increases as the entrainment parameter is increased within the convection scheme of a general circulation model. Our results provide some consistency with this previous work; 6 of the 8 models found here to have positive diagnosed entrainment rates were found by Ahmed and Neelin (2021) to be adequately sensitive to free-tropospheric moisture, while no models with negative diagnosed entrainment rates were found by Ahmed and Neelin (2021) to be adequately sensitive. However, there is no clear relationship between the magnitude of our diagnosed entrainment rate and whether a model was found to be overly or inadequately sensitive; both types of models can be found among the group with small or negative diagnosed entrainment rates.

The pseudo-entrainment rate defined by Emmenegger et al. (2024) is based on similar physical reasoning to our diagnosed entrainment rate, but it depends on only the mean lower-tropospheric stability and humidity in precipitating regions rather than on the spatial and temporal variability of these quantities. Despite this similarity, the relationship between the diagnosed entrainment rate and their pseudo-entrainment rate is weak for the four models included in both studies. Conceptually, one may think of the pseudo-entrainment rate as being determined by the centroid of the histograms in Fig. 3, rather than the full distribution. For many models this would produce a substantially different entrainment value, and this may account for the difference in our diagnostics.

## 4 Implications for extreme weather in a warming climate

We now consider how the stability-humidity relationship changes in a warming climate and the possible implications for two types of extreme weather that are hypothesised to be influenced by convective entrainment. Previous studies have highlighted the importance of entrainment in determining CAPE (Singh and O'Gorman, 2013; Seeley and Romps, 2015) and more recently, extreme wet-bulb temperatures (Duan et al., 2024) in the tropics. Our results may therefore assist in understanding variations in these quantities in climate models, and in particular might help us in understanding the model spread in future projections of the potential for intense thunderstorms and humid heatwaves.

For a given relative humidity, the saturation deficit increases under warming following the Clausius-Clapeyron relation. To the extent that the relative humidity distribution is unchanged in a warmer climate, this results in a shift of the distribution to higher saturation deficit. If the diagnosed entrainment rate also does not change, we would expect models with positive diagnosed entrainment rates to move towards greater values of instability along a slope of constant entrainment. Fig. 6 compares histograms in the stability-humidity phase space in the historical (2000-2014) and SSP5-8.5 (2086-2100) scenarios for each CMIP6 model. In most models, the distribution shifts to both higher saturation deficit and higher instability. In models with positive diagnosed entrainment rates that more closely reproduce reanalysis (Fig. 6a-f), this is seen as a shift roughly aligned with the slope of the historical distribution. In other models, a similar shift occurs, but the relationship to the historical slope is weak, or there is a shift in saturation deficit, but only a weak change in instability. Notably, in the two models discussed previously with strong negative diagnosed entrainment rates, one (IPSL-CM5A-LR) shifts to higher instability in a direction orthogonal to the slope of the historical distribution, while the other (INM-CM5-0) shows weak changes in instability.

In summary, for models with positive diagnosed entrainment rates, the magnitude of the diagnosed entrainment rate appears to partially control the change in instability in a warmer climate. This suggests the diagnosed entrainment rate may provide a useful constraint on future projections.

#### 260 **4.1 CAPE**






CAPE is an important large-scale condition associated with intense thunderstorms and provides an upper limit on buoyancy-driven updraft strength. Climate models robustly project future increases in CAPE (Diffenbaugh et al., 2013; Seeley and Romps, 2015; Chen et al., 2020), but with a large spread in the rate of increase across different model projections (Singh et al., 2017).

According to the ZBP assumption, CAPE results from entrainment acting to deviate the convective lapse rate from that of a moist adiabat. An increase in the saturation deficit and/or the entrainment rate would result in a larger deviation from a moist adiabat and therefore increased CAPE. Indeed, Wing and Singh (2024) attributed much of the variation in CAPE across idealised simulations of radiative-convective equilibrium to variations in convective entrainment.

In a warming climate, the ZBP assumption leads to the expectation that CAPE will increase as the saturation deficit increases.

70 Assuming the entrainment rate and relative humidity remains fixed under warming, this increase occurs at a rate slightly above the Clausius-Clapeyron scaling rate of 7% K<sup>-1</sup> (Romps, 2016; Wing and Singh, 2024). Here, we test these theoretical

**Figure 6.** 2D histograms depicting the relationship between the vertical change in MSE\* and the saturation deficit between 850 hPa and 500 hPa at the beginning of the century (2000-2014, blue) and the end of the century under SSP5-8.5 (2086-2100, red) for the fourteen CMIP6 models. The CMIP6 models are ordered from the strongest negative slope to the strongest positive slope in the historical period. The distribution includes only ocean surface points that exceed the precipitation threshold between 20°N-20°S.

Figure 7. (a) The 95th percentile of CAPE in the historical scenario (2000-2014) for ocean surfaces and precipitating points only plotted against the diagnosed entrainment rate. (b) The fractional increase in the 95th percentile of CAPE at the end of the 21st century (2086-2100) under SSP5-8.5 relative to the start of the 21st century (2000-2014) per unit global warming plotted against the diagnosed entrainment rate in at the start of the 21st century (2000-2014). Error bars represent the structural uncertainty in the diagnosed entrainment rate. The range of diagnosed entrainment rates for ERA5 (dark grey shading) and MERRA2 (light grey shading) are also plotted. The value of the correlation coefficient, r, is calculated across models with positive diagnosed entrainment rates only.

predictions by examining how CAPE varies with the diagnosed entrainment rate  $\epsilon_d$  across CMIP6 models in the current and future climate.

To highlight CAPE extremes most relevant to intense thunderstorm potential, we focus on the 95th percentile of daily CAPE, which we denote CAPE<sup>95</sup>. For each model, we calculate the 95th percentile of CAPE at each grid point on days exceeding the precipitation threshold in the historical period (2000-2014) and average this value over oceans in the region 20°S–20°N.

While the ZBP prediction for CAPE would suggest that it increases with the entrainment rate, in the ensemble of fourteen CMIP6 models, there is only a weak relationship between the diagnosed entrainment rate  $\epsilon_d$  and CAPE<sup>95</sup> (Fig. 7a). Focusing on only models with positive diagnosed entrainment rates, since they display some consistency with the ZBP assumption, the correlation between extreme CAPE and the diagnosed entrainment rate is only 0.02. This lack of correlation is, however, driven primarily by one outlier model (KACE-1-0-G); excluding this model increases the correlation to 0.80. Examining the histogram of KACE-1-0-G in the stability-humidity plane shows that, unlike some other models, it does not approach zero stability for low values of saturation deficit (compare to MRI-ESM2-0, for instance), and this may partially explain its low value of CAPE. However, it is also likely that lower-tropospheric stability is not the only control on CAPE extremes, and CAPE can vary for reasons beyond the effect of entrainment on the lapse rate.

Despite the weak climatological relationship between CAPE extremes and the diagnosed entrainment rate, the fractional increase in CAPE per unit global warming is generally larger in models with a positive diagnosed entrainment rate (9-14%  $K^{-1}$ ) compared to those with a negative diagnosed entrainment rate (3-9%  $K^{-1}$ ) (Fig. 7b). This is physically consistent with the ZBP assumption, which implies increases in CAPE larger than Clausius-Clapeyron scaling, assuming entrainment controls the tropospheric lapse rate even under conditions of large CAPE.

The results for changing CAPE under warming therefore suggest a possible constraint on future projections of CAPE; models that reproduce the correct slope of the stability-humidity relationship have an average increase in CAPE under warming of  $10.3 \% K^{-1}$ , larger than the  $8.4 \% K^{-1}$  average we see for the full ensemble of models used in this study and larger than the ensemble mean of  $9.4 \% K^{-1}$  found for CMIP5 (Singh et al., 2017). However, the presence of KACE-1-0-G, which has a low climatological CAPE<sup>95</sup> value despite its high entrainment rate, suggests that further work is required to fully understand the extent to which entrainment can be thought of as the primary control of CAPE in the tropics.

# 4.2 Humid heatwaves








Humid heatwaves occur under the combination of high temperature and humidity that causes heat stress (Matthews et al., 2025), and their frequency and intensity are projected to increase under warming (Coffel et al., 2018; Rogers et al., 2021). A common measure of the co-occurrence of high temperatures and humidity is the wet-bulb temperature; theories for humid heatwaves seek to determine the maximum possible wet-bulb temperature in the boundary layer before the onset of convection (Zhang et al., 2021; Raymond et al., 2021). Recently, Duan et al. (2024) showed that convective entrainment plays an important role in delaying convection and allowing high wet-bulb temperatures to build up. They found that the effect of entrainment causes values of extreme wet-bulb temperatures in the tropics to be  $\sim 2$  K higher than when entrainment is neglected. Further, they argued that entrainment also causes the rate of increase in wet-bulb temperature per unit of tropical warming to be larger than it otherwise would be, implying that a higher entrainment rate would result in a larger increase in wet-bulb temperature extremes in a warming climate.

The importance of entrainment for humid heatwaves can be understood by noting that wet-bulb temperature is closely related to moist static energy in the boundary layer. Deep convection onset occurs when the boundary-layer moist static energy is high enough that air parcels rising through clouds remain neutrally or positively buoyant up to the mid-troposphere (Ahmed and

Neelin, 2018). Applying the ZBP assumption, this requires that the boundary-layer moist static energy ( $MSE_{BL}$ ) exceeds the mid-tropospheric saturation moist static energy ( $MSE_{MT}^*$ ) by an amount,

$$MSE_{BL} - MSE_{MT}^* = \epsilon \Delta z \ell \overline{(q^* - q)}, \tag{4}$$

where we have simply evaluated (2) from the lifted condensation level to the mid-troposphere. Since the mid-tropospheric saturation moist static energy varies little spatially (Bao et al., 2022), the left-hand side of the above equation depends largely on properties of the boundary layer, and represents a measure of wet-bulb temperature anomalies relative to the mean tropics. The ZBP assumption therefore tells us that the magnitude of humid heatwaves is controlled by the right-hand side of (4); it increases with entrainment and the saturation deficit. To test this hypothesis, we investigate whether differences in the diagnosed entrainment rate across models have implications for the projection of extremes of boundary-layer moist static energy, and therefore humid heatwaves.

We follow Duan et al. (2024) and take the exceedance of the zonal-mean MSE\* at 500 hPa by the near-surface MSE, referred to as boundary-layer instability, as our measure of the co-occurrence of high humidity and temperatures. This provides a statistic that has a clear connection to (4), but that has also been shown to be related to the near-surface wet-bulb temperature that is commonly used to measure humid heatwaves. To define a measure of the maximum boundary layer instability, we calculate the difference between the daily-mean MSE calculated from the 2 m temperature and humidity and the zonal- and daily-mean MSE\* at 500 hPa averaged over the top 1% of daily near-surface MSE values in the historical period (2000-2014) over oceans only. These boundary-layer instability values are then plotted against the diagnosed entrainment rates of each model (Fig. 8a). There is a strong positive correlation for models with positive diagnosed entrainment rates, indicating that the amount of moist enthalpy that can build up at the surface increases with the magnitude of the entrainment rate. This result is consistent with the argument presented in (4) and is intuitively sensible, as greater mixing of dry environmental air would further delay the onset of convection.

Under warming, (4) becomes,







$$\Delta(\text{MSE}_{\text{BL}} - \text{MSE}_{\text{MT}}^*) = \epsilon \Delta z \ell \Delta \overline{(q^* - q)}, \tag{5}$$

assuming the entrainment rate does not vary with warming. Therefore, (5) predicts an increase in the boundary-layer instability that scales with the saturation deficit, implying, under fixed relative humidity, an increase in the boundary-layer instability following the Clausius-Clapeyron scaling. Fig. 8b plots the fractional increase in the boundary-layer instability in the CMIP6 ensemble per degree of global warming. The models project increases of boundary-layer instability broadly consistent with the expectation from the ZBP assumption, but with a large range between  $\sim 4$ -10% K<sup>-1</sup>. While there are somewhat higher increases in boundary layer instability for models with positive diagnosed entrainment rates than those with small or negative ones, this difference is not as pronounced as for projections of CAPE<sup>95</sup>. Interestingly, however, for the models with positive diagnosed entrainment rates, this increase in boundary-layer instability is correlated with  $\epsilon_d$  (r=0.59). That is, there are generally greater increases in boundary-layer instability for those models with larger  $\epsilon_d$  values.

The above result suggests a possible constraint on humid heatwave projections based on the diagnosed entrainment rate and the subset of models with positive entrainment rates. Since the diagnosed entrainment rate from reanalyses is on the upper end

**Figure 8.** (a) The average boundary-layer instability for the top 1% of daily MSE values in the historical scenario (2000-2014) for ocean surfaces plotted against the diagnosed entrainment rate for each model. (b) The fractional increase in average boundary layer instability for the top 1% of daily MSE values at the end of the century (2086-2100) under SSP5-8.5 relative to the start of the century (2000-2014) per unit global warming plotted against the diagnosed entrainment rate. Error bars represent the structural uncertainty in the diagnosed entrainment rate. The range of diagnosed entrainment rates for ERA5 (dark grey shading) and MERRA2 (light grey shading) are also plotted. Values of the correlation coefficient, r, are calculated for models with positive diagnosed entrainment rates only.

of model distribution, the constraint would suggest a change in boundary-layer instability at the upper end of that predicted by the model ensemble. However, we note that the correlation between  $\epsilon_d$  and fractional increases in boundary-layer instability is

not a direct prediction of the ZBP assumption. Rather, this assumption would suggest increases following Clausius Clapeyron for all models, if the entrainment rate is constant with warming. However, we see a range of increases in the models outside of the 6-7% increase implied by Clausius-Clapeyron scaling, suggesting that relative humidity or the entrainment rate may not be constant under warming. Alternatively, or additionally, it may suggest that there are other influences on the increase of boundary layer instability other than what is implied by the ZBP assumption. This, coupled with the large model scatter, gives us low confidence in any emergent constraint based on this relationship. Nevertheless, our results here and in the previous section provide some encouragement that  $\epsilon_d$  provides some information about model projections of quantities affected by convective entrainment. Developing further methods to use process-oriented diagnostics to constrain projections of both humid heatwaves and CAPE extremes in a warming climate is therefore a promising direction for future work.

#### 5 Discussion and Conclusions







In this study we have applied the relationship between instability and humidity, recently argued to be an indicator of convective mixing (Palmer and Singh, 2024), as a process-oriented diagnostic for state-of-the-art climate models. While many of the fourteen models analysed reproduce the dependence of instability on humidity in convecting regions seen in observations and reanalyses, some models have weak or even opposite relationships. The definite cause of these differences in the relationship is difficult to ascertain without detailed mechanism denial experiments, nevertheless, we hypothesise that it may relate to structural differences in each model's convective parameterisation. For example, models that do not include a plume-based mixing scheme cannot reproduce the observed stability-humidity relationship. This appears to be the reason that two models (INM-CM5-0 & IPSL-CM6A-LR) produce strongly negative diagnosed entrainment rates. Further, we find that nearly all models that reproduce a positive diagnosed entrainment rate use a convective closure based on the removal of dilute CAPE over a specified timescale. This is physically consistent with the ZBP assumption, which requires convection to rapidly relax the atmosphere toward neutrality with respect to an entraining plume. This provides some support for the validity of these CAPE closures, although we note that the example of MPI-ESM1-2 shows that using such a closure is not a sufficient condition for producing a positive diagnosed entrainment rate.

In principle, the diagnosed entrainment rate may be compared to that of reanalyses in order to evaluate climate models, as has recently been attempted using similar diagnostics derived for the sensitivity of precipitation to humidity (Ahmed and Neelin, 2021; Emmenegger et al., 2024). However, we note that reanalyses themselves may be affected by the convection scheme of the analysis model, and it is therefore difficult to provide an observational uncertainty on the observed stability-humidity relationship. Radiosonde observations appear to confirm a positive value of the diagnosed entrainment rate (Palmer and Singh, 2024), but further work is needed to provide a more quantitative observational bound on its value.

We also explored the applicability of the stability-humidity relationship to projections of extremes, namely CAPE and humid heatwaves. For models with positive diagnosed entrainment rates, there was a relatively strong relationship between  $\epsilon_d$  and historical CAPE extremes, albeit with a notable outlier. Further, models that reproduce the correct sign of the relationship between stability and humidity project larger increases in CAPE with warming. Models with positive diagnosed entrainment

rates generally produce increases in CAPE with warming at or above that of the Clausius-Clapeyron relation, consistent with theoretical expectations based on the ZBP assumption (Romps, 2016; Wing and Singh, 2024), whereas models with negative diagnosed entrainment rates produce weaker increases.

If one argues that models that cannot reproduce the sign of the observed humidity-stability relationship should be discounted, the above results imply that increases in CAPE under warming are likely to be greater than the ensemble mean of CMIP6 models in this study and the ensemble mean of models in CMIP5 used by Singh et al. (2017). However, the lack of correlation between CAPE and the diagnosed entrainment rate in the historical climatology implies that CAPE is not a simple function of entrainment as suggested by the ZBP assumption, and further work is needed to understand the different factors controlling CAPE before a strong constraint on its future projections can be produced.






A strong correlation across models was found between the diagnosed entrainment rate and boundary-layer instability on days of extreme moist static energy—our index for humid heatwaves. This is consistent with the ZBP assumption and with Duan et al. (2024); stronger mixing in the lower troposphere suggests greater instability and ability to build up moist enthalpy at the surface before convection is triggered. Under warming, increases in boundary-layer instability were larger for models with higher entrainment rates, but there was not a strong separation between models with robustly positive diagnosed entrainment rates and those with small or negative values. This points toward a possible method for constraining humid heatwave projections, but suggests that humid heatwaves are also sensitive to other factors beyond the diagnosed entrainment rate.

Following Palmer and Singh (2024), we have interpreted the relationship between stability and humidity in the tropical troposphere through the lens of convective entrainment. This interpretation has theoretical support from arguments based on the ZBP assumption, and the relationship itself has been found to be consistent among different reanalyses and a similar, albeit noisy, relationship between instability and humidity has been found in radiosonde soundings (Palmer and Singh, 2024). Moreover, our finding that CMIP6 models produce a wide range of stability-humidity relationships suggests that this relationship is a result of uncertain convective physics, of which entrainment and mixing are prime candidates. Nevertheless, other explanations for this relationship are possible, and large-scale dynamics are likely to play a role in determining the degree to which local versus remote influences control the lapse rate in convecting regions (Bao et al., 2022). Further work using storm-resolving models could be useful to tease out these influences without the uncertain aspects of a convection scheme. Additionally, parameter perturbation experiments to vary the entrainment rate within a model's convection scheme could be used to disentangle the direct effect of convective entrainment from other influences, which may include resolved and subgrid-scale vertical mixing.

Code and data availability. The European Centre for Medium-Range Weather Forecasts ERA5 reanalysis dataset is available at https://cds.climate.copernicus.eu/datasets/reanalysis-era5-complete?tab=overview. The Modern-Era Retrospective analysis for Research and Application, Version 2 (MERRA2) reanalysis dataset is available at https://disc.gsfc.nasa.gov/datasets?project=MERRA-2. The CMIP6 model output is available from the Earth-System Grid Federation at https://pcmdi.llnl.gov/CMIP6/. The software library for OpenCV is available at https://github.com/itseez/opencv. Code used for the analysis can be found at https://github.com/lpalmer111/PalmerSingh2025.

Author contributions. LAP did the analysis. Both authors conceptualised the study and wrote the paper.

Competing interests. MSS is a member of the editorial board of Weather and Climate Dynamics

Acknowledgements. We thank three anonymous reviewers for their evaluations of this manuscript. We acknowledge the European Centre for Medium-Range Weather Forecasts, which is responsible for ERA5, the National Aeronautics and Space Administration, which is responsible for MERRA2, and the World Climate Research Programme and the climate modelling groups for producing and making available the CMIP6 output. We acknowledge support from the Australian Research Council and the Centre of Excellence for Weather in the 21st Century through grant nos. CE170100023 and DP230102077, as well as computational resources and services from the National Computational Infrastructure, all funded by the Australian Government. LAP also acknowledges support from the UKRI Frontier Research Guarantee scheme (grant number EP/Y027868/1).

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
