# Peer review of "Entrainment and the tropical tropospheric thermal structure in global climate models"

_EGUsphere, 2025_

## Referee Comment (RC2)

This is a well written and well reasoned manuscript that i found novel and interesting. I particularly like looking at what models do rather than what they purport to do, and this is an example of how we should evaluate what models do to determine if they are adequate for a given purpose. From this point of view if we think that the atmosphere does, in rainy regions, show a robust behavior that a model does not follow, then this is a good reason not to use that model for things that might depend on this relationship. The next step is, of course, to see what depends on this relationship, something the manuscript also takes up.

I have a couple of major comments that the authors may want to consider, but given that the review is part of the public record, they don't need to be addressed as a matter of publication. However, some of the points about editing (particularly as they are part of the ACP standard) should be addressed.

Major comments:

It is not clear to me what controls the spread within the Palmer-Singh space and what assumptions are behind this spread being aligned along a constant $\epsilon$. For one, $\epsilon$ could vary based on environmental conditions, this is certainly the case for Nordeng-Tiedtke which has a small and large-scale entrainment, and even for a fixed entrainment it seems that the adjustment timescale might influence the joint histograms. Given this, it seems that the direction of spread is not necessarily a measure of goodness of fit. It could be argued that IPSL is well described by the entrainment rate that passes through the center of mass of points, something that is supported by the shift with warming, but that variability is expressed by $\epsilon$ varying with environmental conditions, as claimed for MPI-M with Nodeng-Tiedtke.

I don't really agree with the premise that all models are wrong, but in precisely the way that would allow their errors to be corrected by the 'emergent constraint' or 'process oriented diagnostic' . I also would refrain from introducing/using shorthand for things like POD as this gives them an air of precision that is not warranted. I would further maintain that emergent constraints have been largely a dead end that somehow encourages the fantasy that the correct answer is to be found in the garbage heap of inadequate model output, which is used more because it is available, and less because it is demonstrably adequate for purpose. On the other hand the authors's analysis is a very nice way to think about how the world works, in which context more focus on that point, and what statements about the world different models might adequately test, would be more useful. This is actually what the manuscript does in the second part, albeit burdened by the baggage of false ideology (just to phrase things colorfully)

Some minor points, some of which are particular instances of the above are provided below.

• See the ACP typesetting rules, which revers to the IUPAC standards, i.e., roman versus italic and when (operators, name subscripts, are roman) e.g., §1.3 of the GreenBook. Also I persist in trying to encourage my colleagues not to use the word heat as a noun, and not to call enthalpy heat, and not to use capitals for specific quantities… hence $\ell$ for the vaporization enthalpy. The subscript 'v' is not even necessary as the fusion enthalpy does not enter into any of the arguments.

• Given the major comment above, and that the relationship does measure something, for some quantities (i.e., the CAPE estimation) maybe one need not refer to only those models with $\epsilon > 0$. In particularly if there is another process that gives an offset, then the model may well follow the ideas in the ZBP modulo an offset. On the other hand for the boundary layer humidity, it would seem that only models with $\epsilon > 0$ make sense to analyze.

- Although it is well caveated, I'm a bit reluctant to read too much into the reanalyses. Their differences were not insubstantial given the similarities of models and approaches they employ. As more an more large local-area-sounding data sets become available (GATE, FGGE, EUREC4A, OTREC, ORCESTRA and so on), I think this framework could be more usefully employed outside of model space and also outside of the idiosyncratic nature of mostly land based tropical sondes used for routine weather observations.

- To show how little attention I paid to type editing, the only comment here is the preposition 'in' on line 276, maybe it should be 'to'… english is not my mother's tongue and my skills are deteriorating. .. well also maybe line 166 *markedly* would work better than *wildly*. Colorful language works better in reviews.

- On line 110 it can also be said that the entrapment rate used i models is often not as physical as they purport. The values rather express an imposed constraint on precisely the quantities the manuscript diagnoses.

- I had the impression that fewer color steps… i.e., a half dozen or less, and a logarithmic spacing of the histograms would make the shape of the distributions somewhat more clear.

- Finally (line 100) a manuscript that doesn't say something is out of scope, but rather says what is in scope by virtue of which other things are out of scope. Kudos

---

## Author Response (AR1)

**REVIEWER 1**

We thank the reviewer for their constructive feedback and their generous appraisal of the manuscript. Their comments are shown in bold and our responses in italics.

**On Equation (2):**

I may be missing something, but I don't really follow how you get Equation (2). If I was to integrate Eq. (1) between two pressure levels, I would have:

$$dh*/dz = -\varepsilon L(q*-q)$$

$$\int (dh^*/dz) dz = -\varepsilon L \int (q^*-q) dz = -\varepsilon L \int (q^*-q) ((-R_a T)/(pg)) dp$$

where the second equality comes from hydrostatic balance & the ideal gas law. I'm not entirely sure how you get from this to the right hand side of Eq. (2). A few more steps would be appreciated.

Thanks for pointing this out, we have added some extra information at lines 78-85 of the revised manuscript. Essentially, the confusion is that we were keeping the integral in height coordinates. This has been clarified.

**Assorted clarifications/comments**

**Are you using daily data for CMIP6? Could the diurnal cycle be playing a role?**

Yes, we are using daily data for CMIP6 and daily mean fields computed from hourly data for the reanalyses. We acknowledge that using daily averages may neglect important features of the diurnal cycle. However, our focus in this manuscript was capturing larger-scale relationships. Additionally, our analysis focuses on the troposphere over the ocean, where the diurnal cycle of convection is weaker compared to land. We have included some extra text noting the possible importance of the diurnal cycle for the stability-humidity relationship over land at line 101.

L110, when introducing the axis ratio, it would be helpful to state that a 'stronger relationship'='larger axis ratio'

We thank the reviewer for highlighting this. We have added this statement at line 115 in the revised manuscript.

For the scatter plots, please remove the lines behind the markers in the legend key, and make the markers bigger (it's difficult to read them at the moment). For the zero-entrainment lines in the scatter plots, please use "zorder=-10" to put that line

**behind the scatter points. Also, could you please put the Pearson correlation coefficient/p-value in all scatter plots?**

We have updated the scatter plots so that the legend key and the markers are now clearer. The Pearson correlation coefficient is now present in all scatter plots.

**It would be more intuitive to flip the axes in Figures 6 and 7.**

Yes, we agree. We have now updated these figures.

Is there a way to measure the uncertainty in your POD of convective entrainment? For example, the CCCma has negative  $\epsilon_d$  but I imagine this is not statistically different from zero (eyeballing the pdf in Fig 3)? I have a similar skepticism of MPI-ESM-LR's/MIROC6's  $\epsilon_d$  values. Could you bootstrap the slope estimates and give some measure of the uncertainty that way?

We agree that some measure of uncertainty in our POD is necessary. We have now added a new figure (Fig. 4) to more clearly establish the differences in the diagnosed entrainment rate and convective schemes between models. In the manuscript we take the diagnosed entrainment rate based on the contour surrounding 75% of the data. We have now calculated the structural uncertainty of the diagnosed entrainment rate by calculating this value for contours ranging from 30% to 90% of the data. There is some variation in the diagnosed entrainment rates depending on the contour used and the range of entrainment rates can be viewed in Fig 4.

We also investigated the sampling uncertainty in the diagnosed entrainment rate by estimating the slope using different samples of a few years rather than the entire period. There is some variation in the resulting entrainment rate, but this is small compared to the structural uncertainty.

**REVIEWER 2**

We thank the reviewer for their constructive feedback and insight. Their comments are shown in bold and our responses in italics.

**Major comments:**

It is not clear to me what controls the spread within the Palmer-Singh space and what assumptions are behind this spread being aligned along a constant ε. For one, ε could vary based on environmental conditions, this is certainly the case for Nordeng-Tiedtke which has a small and large-scale entrainment, and even for a fixed entrainment it seems that the adjustment timescale might influence the joint histograms. Given this, it seems that the direction of spread is not necessarily a measure of goodness of fit. It could be argued that IPSL is well described by the entrainment rate that passes through the center of mass of points, something that is supported by the shift with warming, but that variability is expressed by varying with environmental conditions, as claimed for MPI-M with Nodeng-Tiedtke.

We thank the reviewer for this comment. We acknowledge that a model's entrainment parameter need not be constant, and the diagnosed value  $\epsilon_d$  may be affected by model tuning. We have added a comment noting this at lines 123-127.

It is also true that the centre of mass of points could also lay claim to be a measure of entrainment, and a similar approach has been used by Emmenegger et al. (2024) to estimate convective mixing in climate models. However, we would argue that the mean stability is more easily tuned by varying other aspects of a model, and so the reviewer's argument below that  $\epsilon_d$  may reflect observational constraints already baked in actually points towards using variability rather than the mean state. We note that the centre of mass of the distribution gives a diagnostic more similar to that used in Emmenegger et al. (2024) at line 236.

I don't really agree with the premise that all models are wrong, but in precisely the way that would allow their errors to be corrected by the 'emergent constraint' or 'process oriented diagnostic'. I also would refrain from introducing/using shorthand for things like POD as this gives them an air of precision that is not warranted. I would further maintain that emergent constraints have been largely a dead end that somehow encourages the fantasy that the correct answer is to be

found in the garbage heap of inadequate model output, which is used more because it is available, and less because it is demonstrably adequate for purpose. On the other hand the authors's analysis is a very nice way to think about how the world works, in which context more focus on that point, and what statements about the world different models might adequately test, would be more useful. This is actually what the manuscript does in the second part, albeit burdened by the baggage of false ideology (just to phrase things colorfully)

We have removed the acronym "POD" from the revised manuscript, and we have tried to focus the discussion more on statements we can make about the model's behaviour rather than whether they reproduce a given metric (e.g., line 125).

The reviewer's point on emergent constraints is well made, and similar thinking underpins our own reluctance to claim a strong constraint from our results. For example, we remain unconvinced that correlations across models with negative diagnosed entrainment rates are meaningful, and we do not argue that a positive correlation of boundary-layer instability with  $\epsilon_d$  should be taken as a constraint, as it is not a direct prediction of the ZBP assumption.

We have altered the discussion in section 4 (e.g., line 280, line 350) to clarify these points.

Some minor points, some of which are particular instances of the above are provided below.

• See the ACP typesetting rules, which revers to the IUPAC standards, i.e., roman versus italic and when (operators, name subscripts, are roman) e.g., §1.3 of the GreenBook. Also I persist in trying to encourage my colleagues not to use the word heat as a noun, and not to call enthalpy heat, and not to use capitals for specific quantities... hence ℓ for the vaporization enthalpy. The subscript 'v' is not even necessary as the fusion enthalpy does not enter into any of the arguments.

Thank you for this comment. We have now revised our typesetting so it is in line with the ACP typesetting rules. We have also modified the manuscript to avoid using heat as a noun.

Given the major comment above, and that the relationship does measure something, for some quantities (i.e., the CAPE estimation) maybe one need not refer to only those models with ε > 0. In particularly if there is another process that gives an offset, then the model may well follow the ideas in the ZBP modulo an offset. On the other hand for the boundary layer humidity, it would seem that only models with ε > 0 make sense to analyze.

As pointed out by another reviewer, the correlation between CAPE and  $\epsilon_d$  is actually quite high for models with positive  $\epsilon_d$  except for one model. We have therefore reassessed our confidence of the constraint on CAPE and become a little more positive in our language. The higher increase in CAPE for positive  $\epsilon_d$  models under warming does provide some evidence that CAPE increases will be on the higher end of the model distribution.

Having said that, we are still skeptical of interpreting models with  $\epsilon_d$ <0 in physical terms. The offset argument would make sense if we were estimating entrainment using the centre of mass of the stability-humidity phase space. But it is harder for us to fathom the processes that would lead to an offset in the slope of this distribution. We therefore continue to focus on the models with  $\epsilon_d$ >0, with the main possibility of a constraint on behaviour coming from a downweighing of these models in projections.

Although it is well caveated, I'm a bit reluctant to read too much into the
reanalyses. Their differences were not insubstantial given the similarities of
models and approaches they employ. As more and more large local-areasounding data sets become available (GATE, FGGE, EUREC4A, OTREC,
ORCESTRA and so on), I think this framework could be more usefully
employed outside of model space and also outside of the idiosyncratic
nature of mostly land based tropical sondes used for routine weather
observations.

This is a valid point, and we recognise the reviewer's reluctance. We hope that this framework can be used with other datasets in the future.

 To show how little attention I paid to type editing, the only comment here is the preposition 'in' on line 276, maybe it should be 'to'... english is not my mother's tongue and my skills are deteriorating. .. well also maybe line 166 markedly would work better than wildly. Colorful language works better in reviews.

Thank you, we have updated these in the revised manuscript.

On line 110 it can also be said that the entrapment rate used i models is
often not as physical as they purport. The values rather express an imposed
constraint on precisely the quantities the manuscript diagnoses.

Thank you for this comment. We have updated our manuscript to reflect this point (lines 124-127).

• I had the impression that fewer color steps... i.e., a half dozen or less, and a logarithmic spacing of the histograms would make the shape of the distributions somewhat more clear.

We have created these plots in the past using a log scale; however these plots were very noisy and we don't believe they best captured the distribution. We experimented with fewer colour steps in the 2D histograms, however we concluded that the original 2D histograms best communicate our results.

**REVIEWER 3**

We thank the reviewer for their constructive feedback and their insight. Their comments are shown in bold and our responses in italics.

I am a little confused about how the authors interpret their scatter plots, esp. figs 4 and 6. In both cases low correlations are reported for models with positive diagnosed entrainment rate, and this seems to strongly influence the authors' interpretation, as they express low confidence in the resulting emergent constraints. However in both cases it looks to me like there is just a single outlier, without which the correlation would be much higher. Am I misunderstanding something? If not, this seems worth discussing explicitly. The choice of which models to include or not include in a study like this is somewhat arbitrary, and in fact which models exist and how they are constructed is also arbitrary. It seems a bit strange to let the overall conclusions be so strongly weakened by one flaky model.

We have now included a comment to highlight that the correlation increases when excluding the model (KACE-1-0-G) this reviewer references, and we have adjusted our conclusions slightly to be a bit more positive about the emergent constraint. Accounting for the structural uncertainty in our method of diagnosing the entrainment rate, this model does lie outside the range of the reanalyses, and in that sense may be an outlier. However, we still believe this model should conform to the ZBP model due to its positive entrainment rate. Therefore, we remain cautious to avoid overstating conclusions.

Relatedly, starting line 319: "However we note that... is not predicted by the ZBP assumption...." This is not obvious and, since it seems important to the conclusions, I suggest spelling out a bit more why this is the case (I assume the essence could be communicated in a couple of lines with perhaps an equation). It's confusing in any case because then why plot things vs. entrainment rate? In figs 4 and 6 the correlation is reported as weak (notwithstanding this is mostly due to one outlier, I think, per above) and so the emergent constraint is weak. Here, the correlation is strong, and yet still the emergent constraint is reported as weak. So why make this plot (fig. 7)? Is there a different plot that could have been made where a strong correlation would have been more meaningful?

Thanks for this comment. We have revised the section to be clearer, and we have included an additional equation. Essentially, the ZBP assumption predicts an increase in boundary layer instability following the Clausius-Clapeyron equation, which doesn't

depend on the entrainment rate. Plotting the change in boundary layer instability against entrainment rate was firstly to highlight differences between models with positive and negative diagnosed entrainment rates, as was also highlighted in the CAPE section. We have adjusted this paragraph to clarify this (lines 332-336).

Notwithstanding the above, it is interesting that there is a correlation with entrainment, which could arise from relationships between the climatological entrainment rate and changes in entrainment or saturation deficit with warming. However, given we have no theoretical reason to expect such a relationship, we are cautious in using this relationship for an emergent constraint, and we are much more comfortable down weighting models that have negative diagnosed entrainment rates in the historical period.